# Forensic Analysis and Genetic Structure Construction of Chinese Chongming Island Han Based on Y Chromosome STRs and SNPs

**DOI:** 10.3390/genes13081363

**Published:** 2022-07-29

**Authors:** Xiao Zhang, Zhen Tang, Bin Wang, Xindao Zhou, Limin Zhou, Gongying Zhang, Junzhe Tian, Yiqi Zhao, Zhiqing Yao, Lu Tian, Suhua Zhang, Hao Xia, Li Jin, Chengtao Li, Shilin Li

**Affiliations:** 1Department of Anthropology and Human Genetics, School of Life Sciences, Fudan University, Shanghai 200438, China; 19210700130@fudan.edu.cn (X.Z.); 21110700125@m.fudan.edu.cn (Z.T.); 19110700115@fudan.edu.cn (J.T.); 21210700055@m.fudan.edu.cn (Y.Z.); lijin@fudan.edu.cn (L.J.); 2Human Phenome Institute, Fudan University, Shanghai 200438, China; 3Criminal Investigation Department of Shanghai, Pudong New Area, Shanghai 200120, China; wangbin8712@aliyun.com (B.W.); ellen870920zj@163.com (G.Z.); 13386287067@163.com (Z.Y.); 13386281130@189.cn (L.T.); 4Criminal Investigation Department of Shanghai, Chongming Island, Shanghai 202150, China; zxd602821371@163.com (X.Z.); shcmzlm@163.com (L.Z.); 5Academy of Forensic Science, Ministry of Justice, Shanghai 200063, China; zsh-daisy@163.com; 6School of Mathematical Sciences, Tongji University, Shanghai 200092, China; 1950384@tongji.edu.cn

**Keywords:** Y-STR, Y-SNP, Chongming island, population differentiation

## Abstract

Y-chromosome short tandem repeat (Y-STR) and Y-chromosome single nucleotide polymorphism (Y-SNP) are genetic markers on the male Y chromosome for individual identification, forensic applications, and paternal genetic history analysis. In this study we successfully genotyped 38 Y-STR loci and 24 Y-SNP loci of Pudong Han (*n* = 689) and Chongming Han (*n* = 530) in Shanghai. The haplotype diversity of the Y filer platinum genotyping system was the highest in the Han population in the Pudong area of Shanghai (0.99996) and Chongming Island (0.99997). The proportion of unique haplotypes was 97.10% (Pudong) and 98.49% (Chongming), respectively. The multidimensional scaling analysis and phylogenetic analysis were performed according to the genetic distance Rst, which was calculated based on the Y-STR gene frequency data. Moreover, we made a comparison on the frequency distribution analysis and principal component analysis of haplogroups in both populations. As a result, Shanghai Pudong Han, Chongming Island Han, and Jiangsu Han were determined to have a strong genetic affinity. The haplogroup distribution characteristics of the Pudong Han and Chongming Han populations were similar to those of the southern Han population. The results of haplotype network analysis showed that Jiangsu Wujiang Han and Jiangsu Changshu Han had more paternal genetic contributions to the formation of Shanghai Pudong Han and Chongming Island Han. Through the joint analysis of SNPs and STRs, this study deeply analyzed the paternal genetic structure of the Pudong Han and Chongming Han populations. The addition of Y-SNP haplogroups to forensic applications can provide information for pedigree investigation.

## 1. Introduction

The male-specific Y chromosome is an ideal tool for genealogical research, forensic application, and patrilineal immigration research. In addition, 95% of the regions that cannot be exchanged or recombined with the X chromosome are called nonrecombining regions (NRY) [1]. There are many kinds of genetic markers on the Y chromosome, of which the two most studied are Y chromosome short tandem repeat (Y-STR) and Y chromosome single nucleotide polymorphisms (Y-SNP). As a chromosomal marker that can be stably inherited in male families, the Y-STR locus has been used for paternity testing and individual identification for some time. Y chromosome single nucleotide polymorphisms (Y-SNPs) of the nonrecombining portion of the Y chromosome (NRY) play an important role in patrilineal traces of the population. Currently, Y-SNPs are being investigated as genetic markers for pedigree mapping in forensic cases. The mutation rate of Y-STR is 3.78 × 10^−4^~7.44 × 10^−2^ mutation/generation, and the mutation rate of Y-SNP is 1 × 10^−9^ mutation/generation [2,3]. Y-SNPs have an extremely low mutation rate relative to Y-STRs (approximately 1/30,000,000 of Y-SNPs). Combined analysis of Y-STRs and Y-SNPs can be valuable for the forensic application of population genetic structure construction [4,5].

Chongming Island is an alluvial island at the entrance of the Yangtze River at the eastern end of the Yangtze River Delta. It is the third largest island and the largest alluvial sand island in China. It has been more than 1300 years since Chongming Island appeared to its current scale. According to the seventh census in 2020, the population size of Chongming Island is 637,900, more than 99% of whom are Han Chinese [6].

The Pudong area is located in the east of the Huangpu River in Shanghai, at the entrance of the Yangtze River. With the continuous advancement of Shanghai’s urbanization process, a large number of migrants have gathered in Shanghai. According to the 2020 census, there are 5.68 million floating and living populations in Pudong. Investigating Y-STR genetic structure and Y-SNP lineage mapping of the native population in Pudong can be valuable for the case application of paternal biogeographic ancestry inference. Due to the special geographical location of these two places, the two Han populations have unique genetic backgrounds.

In this study, we analyzed 38 Y-STR loci and 24 Y-SNP genetic markers in Chongming Han and Pudong Han populations. This study evaluated the four most common Y-STR genotyping systems and used forensic parameters such as individual identification probability, matching probability, Y-STR haplotyping system discrimination ability, and haplotype diversity value to evaluate their efficacy. Subsequently, based on the Y-STR and Y-SNP genetic markers, population genetic structure analysis was performed, and the results revealed that the Han people in Chongming Island and Pudong have a close genetic relationship with the Han people in Jiangsu. The Jiangsu Wujiang Han population and Jiangsu Changshu Han population have more paternal genetic contributions to the Pudong population and Chongming Han population. Therefore, in our research, a total of 1219 Han Chinese male samples from Chongming Island and the Pudong area were collected to provide raw data for further research.

## 2. Materials and Methods

This study was approved by the Ethics Committee of Fudan University (code: BE1806; date: 3 March 2018) and was strictly implemented in accordance with the relevant requirements of the Declaration of Helsinki [7].

### 2.1. DNA Sample Preparation

In this study, peripheral blood samples from 1219 unrelated Han Chinese male individuals in the Pudong and Chongming areas of Shanghai were collected, and blood samples were retained in Flinders Technology Associates (FTA) blood sample collection cards (Whatman International Ltd., Maidstone, UK). All male individuals in this study are local people whose families have lived there for at least three generations, and all household registration information was verified by the administrative department. Based on the principle of informed consent, the research subjects in the Chongming area (*n* = 530) and Pudong area (*n* = 689) all signed the informed consent form. The geographic locations of Pudong and Chongming Han populations are marked with stars (shown in Figure 1).

### 2.2. DNA Typing of 38 Y-STR Locus and 24 Y-SNP Markers

In this study, the Yfiler™ Platinum PCR Amplification Kit (Thermo Fisher Scientific, Waltham, MA, USA) [8] was used to analyze 38 Y-STR loci, which are listed in Appendix A. The amplification experiment was completed according to the Y filer™ Platinum PCR Amplification Kit workbook. In this study, direct amplification was used, and the DNA blood card sample was cut into 1 mm^2^ using a Harris micropunch for PCR amplification. Amplification products were obtained using the GeneAmp^®^ PCR System 9700 (Thermo Fisher Scientific, Foster City, CA, USA). The reaction system was 10 μL. PCR amplification products were separated by capillary electrophoresis (CE) using an ABI 3500xL Genetic Analyzer (Thermo Fisher Scientific, Foster City, CA, USA). Typing results were analyzed using GeneMapper ID-X v 1.4 software (Thermo Fisher Scientific, Foster City, CA, USA). DNA 007 (Thermo Fisher Scientific, Foster City, CA, USA) in the kit was used as a positive control.

We analyzed 24 Y-SNP markers using the Y-SNP pedigree marker system (Pedigree Tagging System (Suzhou Microread Genetics, Suzhou, Jiangsu, China)) [9], and the primary haplogroups included E-M96, D-JST021355, N-M231, C-M130, O-P186, I-M170, IJ-M429, K-M9, QR-M45, G-M201, and IJK-M522. Subhaplogroups included D1a1a1-N1, D1a2a-P47, O1a-M119, O1b-M268, O1b2-M176, O2-M122, O2a1-KL1, O2a2-P201, O2a2b-P164, O2a2a1a2-M7, O2a2b1a1-M117, C2-M217, and N1a1-M46. These Y-SNP marker definitions strictly follow the specifications of Y Chromosome Haplotype Reference (YHRD, http://yhrd.org, accessed on 21 April 2021) [10], Y Chromosome Consortium (YCC) [11,12], International Society of Genetic Genealogy (ISOGG, http://isogg.org/tree/index.html, accessed on 21 April 2021), and Phylotree [13].

### 2.3. Data Analysis

#### 2.3.1. Y-STR Analytic Methods

We calculated the allele frequencies and gene diversity (GD) values [14] of 38 loci in Chongming and Pudong populations using the direct calculation method [14,15]. The formula is shown below:$$\mathrm{GD}=\frac{n}{\left(n-1\right)}(1-\sum^{\;}p_{i}^{2})$$
where *n* represents the number of samples, and $p_{i}^{}$ represents the allele frequency.

We selected 14 reference populations to compare the GD values of 27 loci, including Jiangsu Wujiang Han [16], Jiangsu Changshu Han [17], Shanghai Han [18], Liaoning Han [19], and other Han [20]. Other Hans include Heilongjiang Han, Beijing Han, Shanxi Han, Shandong Han, Henan Han, Zhejiang Han, Guangxi Han, Hunan Han, Fujian Han, Guangdong Han, and Jiangxi Han. Box plots were created using Microsoft^®^ Excel software [21].

According to Shannon’s instructions [22], the haplotype match probability (HMP), discrimination capacity (DC), fraction of unique haplotypes (FUH), and haplotype diversity (HD) under the four Y-STR locus typing systems were calculated. The four Y-STR DNA typing systems included the AmpFLSTR^®^ Yfiler™ PCR Amplification Kit (Thermo Fisher Scientific, Waltham, MA, USA) [23], Yfiler™ Plus (Thermo Fisher Scientific, Waltham, MA, USA) [24], PowerPlex^®^ Y23 System (Promega, Madison, WI, USA) [25], and Yfiler™ Platinum PCR Amplification Kit (Thermo Fisher Scientific, Waltham, MA, USA) [8].

A total of 26 reference populations were selected in this study, including Jiangsu Wujiang Han [16], Jiangsu Changshu Han [17], Jiangsu Nantong Han [26,27] (*n* = 1336), Jiangsu Changzhou Han [28] (*n* = 2597), other Jiangsu Han [29] (*n* = 377), Anhui Han [30] (*n* = 3110), Guangdong Han [31] (*n* = 1827), Guangxi Han (*n* = 105), Guizhou Han [32] (*n* = 658), Jiangxi Han (*n* = 1325), Liaoning Han [33] (*n* = 3544), Fujian Han (*n* = 857), Zhejiang Ningbo Han (*n* = 925), Shanghai Han [18] (*n* = 777), Sichuan Han [34] (*n* = 337), Zhejiang Han [35,36] (*n* = 2039), Ningxia Hui [37,38] (*n* = 78), Yunnan Bai (*n* = 133), Hainan Li [39,40] (*n* = 497), Inner Mongolian [41] (*n* = 466), Qinghai Tibetans [42,43] (*n* = 590), Yunnan Yi [44,45] (*n* = 452), Nagasaki, Japanese [46] (*n* = 133), Okinawa Japanese [47] (*n* = 55), South Korean [48] (*n* = 272), and African Americans [49,50] (*n* = 1800). Pairwise genetic distance (Rst) matrices for the 28 reference populations were calculated by the “AMOVA” online tool on the YHRD website (http://yhrd.org, accessed on 21 September 2021). The Rst genetic distance matrix was applied to perform multidimensional scaling plotting (MDS) using the “MASS” package of R-Studio (https://www.r-project.org, accessed on 21 September 2021). In addition, we constructed a neighbor-joining tree (NJ tree) [51] using Molecular Evolutionary Genetics Analysis X (MEGA X) [52].

#### 2.3.2. Y-SNP Analytic Methods

We directly calculated the frequencies of the primary haplogroups C, D, N, O, and QR and the frequencies of 18 subhaplogroups. Based on research on haplogroups of Chinese populations in other studies, we selected 16 reference populations. Since the haplogroup classifications of each reference population were different, we redefined the haplogroup results of these reference populations using the 24 Y-SNP Markers of the Pedigree Tagging System and calculated the haplogroup frequency.

The reference population included Jiangsu Wujiang Han [16], Jiangsu Changshu Han [17], Yunnan Han, Shandong Han-1 [53], other Han [20] (Jiangxi Han, Zhejiang Han, Hunan Han, Guangdong Han, Fujian Han, Guangxi Han, Shandong Han-2, Henan Han, Heilongjiang Han, Shanxi Han, Beijing Han), and Ningxia Hui [38]. Based on the results of haplogroup frequencies, we plotted them to perform a principal component analysis (PCA) using IBM SPSS Statistics 21 (IBM Corporation, Armonk, NY, USA) [54].

#### 2.3.3. Y-STR and Y-SNP Joint Analytic Methods

The analysis was performed using the software Network 10.1 [55] (http://www.fluxus-engineering.com, accessed on 26 December 2021), and the graphs were drawn using the software Network Publisher. We selected 15 Y-STR loci, including DYS19, DYS389b (equivalent to DYS389II-DYS389I) [56], DYS390, DYS391, DYS392, DYS393, DYS437, DYS438, DYS439, DYS448, DYS456, DYS458, DYS635, and YGATA-H4. According to the mutation rate of these single-copy sites as the basis for setting the weight of the Network graph, the weight interval was 1–5, and the lower the mutation rate is, the higher the weight [20].

## 3. Results

For successful acquisition of Y-STR and Y-SNP genotyping results, see Appendix A.

### 3.1. Genetic Diversity and Haplotype Diversity of Y-STRs

#### 3.1.1. Allelic Diversity Analysis

The allelic frequencies and corresponding GD values for each Y-STR locus are listed in Appendix A. A total of 247 alleles were detected at 32 single-copy Y-STR loci in the Han population of Chongming Island, and the allele frequencies ranged from 0.0019 to 0.9623. At the multicopy loci DYS385a/b, DYS387S1a/b, and DYS527a/b, 53, 37, and 30 haplotypes were detected, respectively, and the haplotype frequencies were distributed between 0.0019 and 0.1698. GD was distributed between 0.0731 (DYS645)-0.9401 (DYS385a/b). A total of 250 alleles were detected at 32 single-copy Y-STR loci in the Pudong population, and the allele frequencies ranged from 0.0015 to 0.9550. A total of 67, 51, and 40 haplotypes were detected at the multicopy loci DYS385a/b, DYS387S1a/b, and DYS527a/b, respectively, and the haplotype frequencies ranged from 0.0015 to 0.1553. The gene diversity value (GD) of each locus was distributed between 0.0864 (DYS645) and 0.9514 (DYS385a/b).

In this study, we compared the GD values of 17 Han Chinese populations, including the Shanghai population [18], Liaoning population [19], Jiangsu Wujiang Han [16], Jiangsu Changshu Han [17], and other Chinese populations [20]. Other populations include Heilongjiang Han, Beijing Han, Shanxi Han, Shandong Han, Henan Han, Zhejiang Han, Guangxi Han, Hunan Han, Fujian Han, Guangdong Han, and Jiangxi Han. The comparison results are shown in Figure 2. DYS391 and DYS438 exhibited the lowest rate of polymorphisms in the Han Chinese population. DYS518 and DYS449 were the two most polymorphic single-copy loci in the Han Chinese population. Rapidly mutating Y-STR loci can be used to differentiate male individuals from closely related families. The GD values of DYS645, DYS391, and DYS438 in the Chongming population were 0.0731, 0.4723, and 0.2553, respectively. The GD values of other loci were all above 0.5, and the multicopy loci were all above 0.9. The GD values of the three loci DYS645, DYS391, and DYS438 in the Pudong population were 0.0864, 0.4134, and 0.3603, respectively. The GD values of other loci were all greater than 0.5, and the GD values of multicopy loci were all greater than 0.9. In general, the 38 Y-STR locus detection system exhibited a high degree of genetic polymorphism in the population of Pudong and Chongming, which can provide rich genetic information for population genetic research and forensic applications.

#### 3.1.2. Haplotype Diversity Analysis

Y-STR haplotypes play an important role in the identification of paternity [57], forensic evidence identification [58], and individual identification [59]. Currently, there are many commercial Y-STR DNA typing systems, such as the AmpFLSTR^®^ Yfiler™ PCR Amplification Kit [23], Yfiler™ Plus [24], PowerPlex^®^ Y23 System [25], and Yfiler™ Platinum PCR Amplification Kit [8]. Different Y-STR typing systems use different Y-STR locus panels, which have differential discrimination powers. We evaluated the performance of different typing systems according to haplotype diversity (HD), haplotype matching probability (MP), and discrimination capacity of the Y-STR haplotype typing system (DC). The results are shown in Table 1.

The Y filer platinum genotyping system exhibited the highest haplotype diversity values in the Shanghai Pudong area and Chongming Island Han population, which were 0.99996 (Pudong) and 0.99997 (Chongming), respectively. The proportions of unique haplotypes were 97.10% (Pudong) and 98.49% (Chongming), respectively. The Y filer Platinum was found to have the strongest discriminative ability in the four Y-STR genotyping systems discussed in our assay, with the highest proportion of unique haplotypes.

#### 3.1.3. Variation Analysis

Microvariants are rare alleles at the Y-STR locus. We observed 15 microvariants in the Chongming Han population, including three single-copy loci, i.e., DYS627 (18.2, 19.2, and 21.2), DYS458 (14.1), and DYS518 (37.2 and 38.2), and one multicopy locus, i.e., DYS527a/b (21/22.2). We identified 14 copy number variations (CNVs), including six single-copy loci and two multicopy loci: DYS635 (21/22), DYS627 (21/23), DYS19 (15/16), DYS444 (12/13, 13/14 and 11/12), DYS439 (11/12), DYS481 (25/26), DYS387S1a/b (36/41/42, 37/38/39, and 36/37/38), and DYS527a/b (19/20/21). Furthermore, 30 microvariants were found in Pudong Samples, including four single-copy loci, i.e., DYS627 (18.2, 21.2), DYS458 (14.1), DYS448 (19.2), DYS518 (36.2, 37.2, 38.2, and 39.2), and two multicopy loci, i.e., DYS385a/b (12/17.2) and DYS387S1a/b (39.2/39.2). We observed 14 copy number variations (CNVs): DYS460 (10/11), DYS385a/b (12/13/17/18 and 12/13/18/19), DYS387S1 (38/39/40, 36/37/38, 35/37/38, 36/37/39, 37/39/40, 35/36/38, 34/35/39, 38/40/42, and 34,36,38), and DYS527 (20/21/23, 21/22/23, 22/23/24, 22/24/25, 23/24/25, and 21/23/24). We did not observe a null allele at any locus between the two populations. All Pudong and Chongming samples with variants are listed in Appendix A.

It is worth noting that of the 1219 samples in this study, 22 samples had one “.2” type of microvariant at the DYS518 loci, of which 21 samples were haplotypes in the QR haplogroup. This result is the same as previous studies on the relationship between the “DYS518~.2” allele and haplogroup Q [20,53].

### 3.2. Genetic Affinity Analysis

For the further verification on the genetic structure background of Shanghai Pudong Han and Chongming Han, multidimensional scaling analysis (MDS) can be used to effectively explore similarities and differentiation in the genetic background of different populations. Based on the Rst genetic distance between each population, a corresponding multidimensional scaling analysis plot (Figure 3) was constructed to visualize the genetic structure relationship between 28 populations (initial stress = 0.04649). The results of pairwise genetic distances are shown in Appendix A.

The plot shows that 18 Han Chinese are all clustered in the middle of the graph, and the distribution is relatively close. The six ethnic minorities are distributed around the Han population with a relatively scattered distribution.

Among them, Guangxi Han and Hainan Li are clustered together. As reported, Guangxi Han are a group formed by ethnic minorities, so the Guangxi Han and ethnic minorities are seemed to have a high affinity [60]. The clustering of Guizhou Han and Sichuan Han is more obvious because both Guizhou and Sichuan are located in southwestern China and are adjacent [61]. The closest populations to Chongming Han were Pudong Han, Jiangsu Nantong, Jiangsu Wujiang, and Jiangsu Changshu, and we observed strong affinity among them. The Pudong Han people clustered together with Jiangsu Wujiang, Jiangsu Changshu, and Jiangsu Nantong and had a very close genetic relationship. The two populations in Japan as the out group were also clearly clustered into one cluster.

Compared to the other four reference populations in Jiangsu Province, the Pudong Han population is farther from the Jiangsu Changzhou Han population. This result confirms that there is inner genetic structure between the Jiangsu Han population. Pudong is closer to these Jiangsu populations than Chongming Island populations. The geographical locations of Pudong and Jiangsu are closer, and genetic exchange is more likely to occur. However, the Chongming island population is the result of immigration of the mainland population. Since transportation between the island and the mainland is difficult, this has led to partial geographic isolation, resulting in genetic drift.

To further reveal the genetic structure among populations, we constructed a phylogenetic tree based on the neighbor-joining method (Figure 4). We observed that Han populations were almost all in the upper half of the phylogenetic tree, while ethnic minorities and out groups were in the lower half of the tree. Except for Guangxi Han and Guizhou Han, Chinese Han populations are clustered under one clade. Pudong Han, Chongming Han, and Jiangsu Han are clustered under one branch at the top of the phylogenetic tree. This further indicates the kinship between Pudong Han, Chongming Han, and Jiangsu Han. Despite this, Yunnan Bai and Yunnan Yi are clustered together, because they are all ethnic minorities in Yunnan Province. In out group populations, the Japanese and Korean populations were clustered into one cluster, while the genetic affinity between two Japanese populations is obviously closer than that between Japanese and Korean. Among the Han population, only Guizhou Han and Guangxi Han were not clustered under the same branch as other Han populations. Previous studies have shown that the genetic relationship between Guizhou Han and ethnic minorities is relatively close [62].

### 3.3. Y-SNP Analysis

#### 3.3.1. Y-Chromosomal Haplogroup Distribution

In this study, 24 Y-SNP loci were analyzed, and 18 haplogroups were defined (D, D1a1a1, C, C2, IJ, K, QR, N, N1a1, O1a, O1b, O1b2, O2, O2a1, O2a2, O2a2a1a2, O2a2b, and O2a2b1a1). The distribution of haplogroups within the Pudong and Chongming Han populations is shown in Appendix A. The detailed haplogroup results of the population in the Chongming area were C2-M217 (7.17%), C-M130 (0.19%), D1a1a1-N1 (0.75%), D-JST021355 (0.19%), N-M231 (9.06%), N1a1-M46 (4.53%), O1a-M119 (29.62%), O1b2-M176 (0.19%), O1b-M268 (4.15%), O2a1-KL1 (13.4%), O2a2a1a2-M7 (3.77%), O2a2b1a1- M117 (9.43%), O2a2b-P164 (14.15%), O2a2-P201 (1.13%), O2-M122 (0.75%), and QR-M45 (1.51%). The detailed haplogroup results of the Pudong population were C2--M217 (6.97%), D1a1a1-N1 (1.31%), N-M231 (4.79%), N1a1-M46 (2.47%), O1a-M119 (24.53%), O1b2-M176 (0.15%), O1b-M268 (6.82%), O2a1-KL1 (22.35%), O2a2a1a2-M7 (2.61%), O2a2b1a1-M117 (10.89%), O2a2b-P164 (10.89%), O2a2-P201 (1.89%), O2-M122 (2.18%), and QR-M45 (2.18%) (Figure 5).

Haplogroups worldwide can be divided into more than 20 major groups, numbered A–T, among which C, D, N, and O are the four major haplogroups in East Asia, accounting for approximately 93% of East Asian males [63]. More than 70% of the Chinese Han population belongs to the O haplogroup. The O haplogroup frequencies in Chongming Han and Pudong Han in this study were 77% and 82%, respectively. This result is consistent with the frequency of the overall haplogroup distribution in the Chinese Han population. The O haplogroup can be further divided into two major clades, primarily categorized into the O1 and O2 haplogroups, which account for 60% of East Asian males. There is a large population of Han Chinese in China, and the haplogroup frequency data can reflect the genetic differences between people in different geographical locations. The distribution of the O1 haplogroup is influenced by geographic pattern. The O1 haplogroup has a low frequency distribution in northern provinces, almost all below 20%. O1 is relatively higher in the Han population in southern provinces, all being greater than 25% [64]. In this study, Chongming O1a-M119 accounted for 29.62%, and Pudong O1a-M119 accounted for 24.53%. It was the haplogroup with the highest proportion of these two populations. The O1a-M119 haplogroup is concentrated on the southeastern coast of China, the Dong-Dai population, and the Taiwan aborigines. Shandong Province has always been an area with a low frequency distribution of O1a-M119. In previous studies, the proportion of O1a-M119 in Shandong Han nationality was only 3.1% [20] and 3.0% [53]. The proportions of the O1b-M268 haplogroup in Pudong and Chongming Han were 6.97% and 4.43%, respectively. O1b-M268 is widely distributed in northern Eurasia and is a common haplogroup in southern Han populations. Among the Han people in Pudong and Chongming Island, the O2 haplogroup accounted for 50.81% and 42.63%, respectively. The O2 haplogroup is the predominant haplogroup in East Asian populations. The results of previous studies on O2a2b-P164 and O2a1-KL1 revealed that the average distribution ratio of O2a2b-P164 in the southern Han was 21.54%, and the distribution in the northern Han was 34.11% [65]. O2a2b-P164 reflects differences between the northern and southern populations. In this study, the proportions of the O2a2b-P164 haplogroups in Pudong and Chongming Han were 21.78% and 23.58%, respectively. The results of this frequency distribution are more in line with the characteristics of the southern Han. The O2a1-KL1 haplogroup is widely distributed in northern, southern, and eastern China, and the distribution ratio is approximately 20%, which is relatively evenly distributed throughout the Chinese population. In the Chongming Han population, we observed that the proportion of the N-M231 haplogroup was 13.59%, while the proportion of the N-M231 haplogroup in the Pudong Han population was only 7.26%. The N haplogroup is widely distributed in northern Eurasia. Previous studies have suggested that the high frequency of the N-M231 haplogroup in Eastern Europe is the result of the westward migration of populations from inland Asia [66]. The high frequency distribution of haplogroup N-M231 in Chongming Island may be caused by the “founder effect” of genetic drift. After the initial population of the N-M231 haplogroup settled on the island, due to the small population on the island, the gene frequency of the N-M231 haplogroup population dominated and expanded, resulting in the current frequency distribution.

#### 3.3.2. Principal Component Analysis

To further study the paternal genetic relationship among the populations, we integrated the haplogroup frequencies of 16 other Chinese populations for comparison. Based on the haplogroup frequencies of Pudong Han, Chongming Han, and 16 other reference populations, PCA plot was performed. The plot of the PCA was expressed in the coordinate system of the first two principal components. The results are shown in Figure 6. The first two principal components explained 72.336% of the Y chromosome variation. The first principal component clearly describes the southern population of the Chinese population (Chongming Han, Pudong Han, Jiangsu Wujiang Han, Jiangsu Changshu Han, Hunan Han, Jiangxi Han, Zhejiang Han, Fujian Han, Yunnan Han, Guangdong Han, Guangxi Han) and the geographical pattern of northern populations (Heilongjiang Han, Beijing Han, Shanxi Han, Shandong Han-1, Henan Han, Shandong Han-2, Ningxia Hui). There is an internal genetic structure in the Han population in the north and south Chinese populations [67,68].

The distribution of the southern population in this figure is relatively loose, and the distribution of the northern Han population in the figure is relatively tight. This shows that southern Hans have greater Y chromosome genetic structural variation than northern Hans [20]. The Guangxi Han people appear as outliers in the graph, probably because the Guangxi Pinghua population was formed by aboriginal minorities who embraced Han culture [20,69]. The PCA results demonstrated that Chongming Han, Pudong Han, Jiangsu Wujiang Han, and Jiangsu Changshu Han had a close genetic relationship.

### 3.4. Network Analysis

To discern the genetic structure between Pudong Han, Chongming Han and Jiangsu Han in details. We used the median joining (MJ) method to create the STR haplotype network under the O1a-M119 haplogroup. The network plot was based on 15 Y-STR loci exploring the connection and differentiation relationship of STR haplotypes under the O1a-M119 haplogroup (Figure 7). The confirmation of the ancestral haplotype of O1a-M119 is based on the 1000 Genomes Project III [70,71] Y-SNP and Y-STR datasets in Central and East Asia. The haplotype with the closest genetic distance between a haplotype and other haplotypes is calculated to define the ancestral haplogroup through EA YPredictor [72].

The joint analysis of Y-SNPs and Y-STRs can be used to infer paternal migration routes [73]. The O1a-M119 network showed that Chongming Han and Pudong Han were located downstream of Jiangsu Wujiang Han and Jiangsu Changshu Han, suggesting that Chongming Han and Pudong Han might be immigrants from Jiangsu Han. This result indicates that under the O1a-M119 haplogroup, the Jiangsu Wujiang population and the Jiangsu Changshu population have more paternal genetic contributions to the formation of the Chongming and Pudong Han populations. The haplogroup network exhibits a star-like spread, illustrating the phenomenon of population expansion in the Pudong and Chongming areas. This may be due to genetic drift. According to some historical records, the original source of the Han population in Chongming Island was the Han population in Jiangsu Province [6].

## 4. Conclusions

In this study, we analyzed 38 Y-STR loci and 24 Y-SNP genetic markers in Chongming Han and Pudong Han populations. Genetic diversity analysis was performed based on the results of Y-STR. We evaluated four different Y-STR genotyping systems separately using four forensic parameters. Variation analysis counted microvariants and copy number variations and verified the correlation between the “0.2” mutation of DYS518 and the QR haplogroup. The results of population genetic affinity analysis indicated a paternal genetic correlation among Pudong Han, Chongming Han, and Jiangsu Han. The Y-SNP haplogroup analysis revealed that the primary haplogroup of the two populations was O1a-M119. The proportions were 29.62% (Chongming) and 24.53% (Pudong). The results of the Y-SNP haplogroup frequencies of the two populations revealed that the haplogroup characteristics were closer to those of the southern Han. In addition, under the O1a-M119 haplogroup, the results of the haplotype network analysis suggested a paternal genetic contribution relationship between Jiangsu Wujiang Han and Jiangsu Changshu Han to Chongming Han and Pudong Han. This study explored the genetic structure of the two populations from the perspective of molecular biology and compared them to other Han populations in China to determine the correlations and differences in the genetic structure between these populations. The results of this chapter provide new ideas for the gradual refinement of population research on the Han population in China and provide original population data for the practical application of forensic medicine.

## Figures and Tables

**Figure 1 genes-13-01363-f001:**
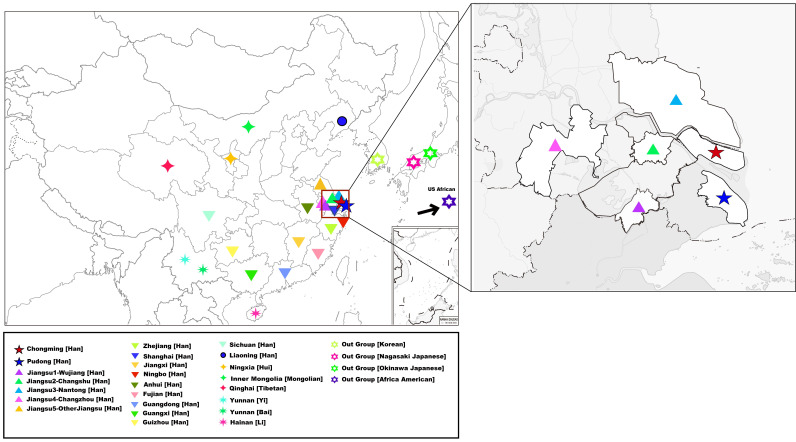
Geographical positions of the Pudong Han and Chongming Han populations with other 26reference populations in this study.

**Figure 2 genes-13-01363-f002:**
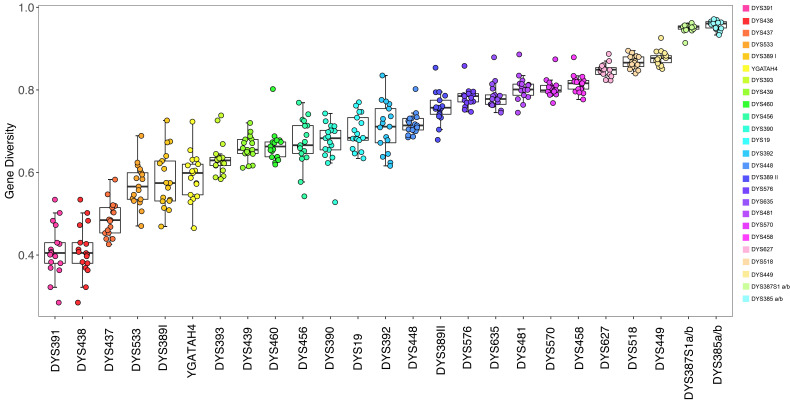
Boxplots comparing GD values in 17 populations based on 27 common Y-STR loci.

**Figure 3 genes-13-01363-f003:**
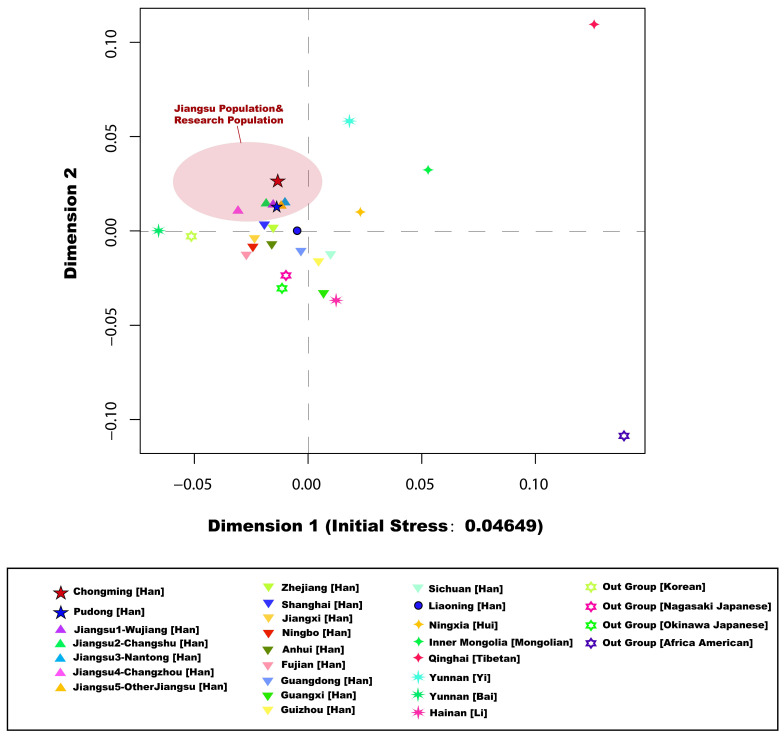
Multidimensional scaling (MDS) plots of 28 populations based on pairwise genetic distances (Rst). Five populations of Han nationality in Jiangsu Province and the studied population are shown in the red circle.

**Figure 4 genes-13-01363-f004:**
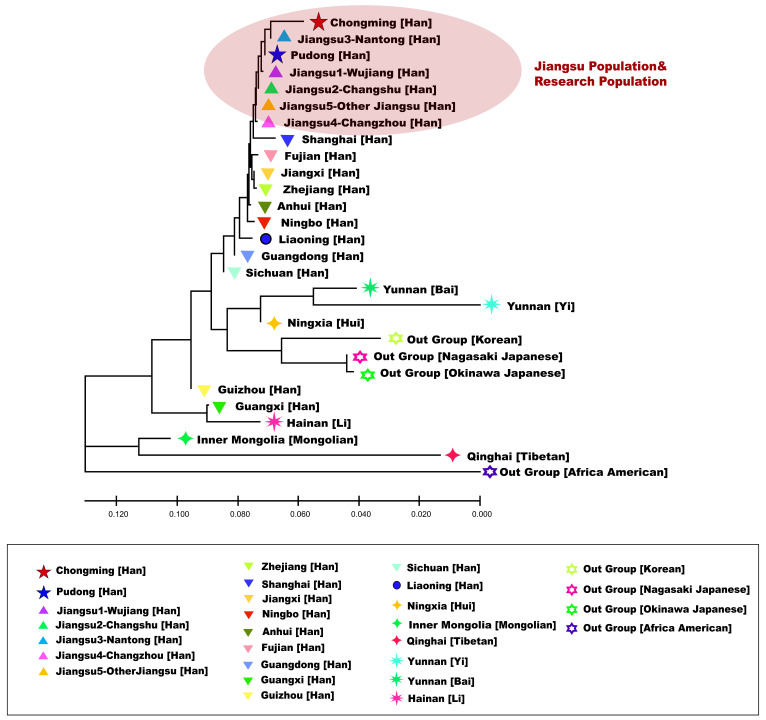
Phylogenetic tree of 28 populations constructed using the neighbor-joining method. Five populations of Han nationality in Jiangsu Province and the studied population are shown in the red circle. The illustrations correspond to those shown in Figure 1 and Figure 3.

**Figure 5 genes-13-01363-f005:**
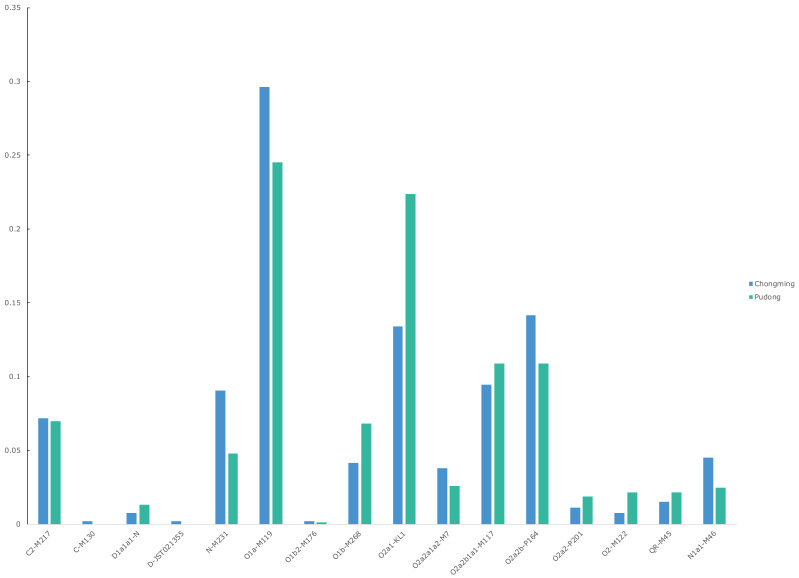
Detailed haplogroup distribution of the two populations in Chongming and Pudong.

**Figure 6 genes-13-01363-f006:**
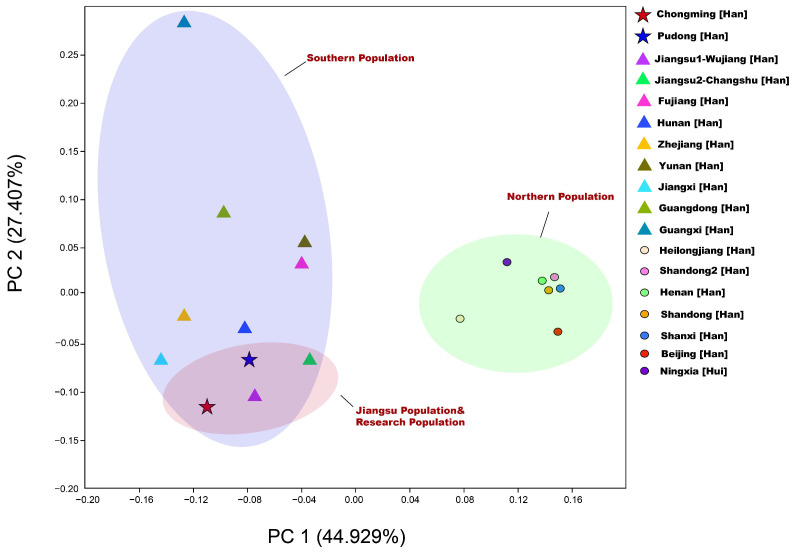
Principal component analysis based on the frequencies of haplogroups.

**Figure 7 genes-13-01363-f007:**
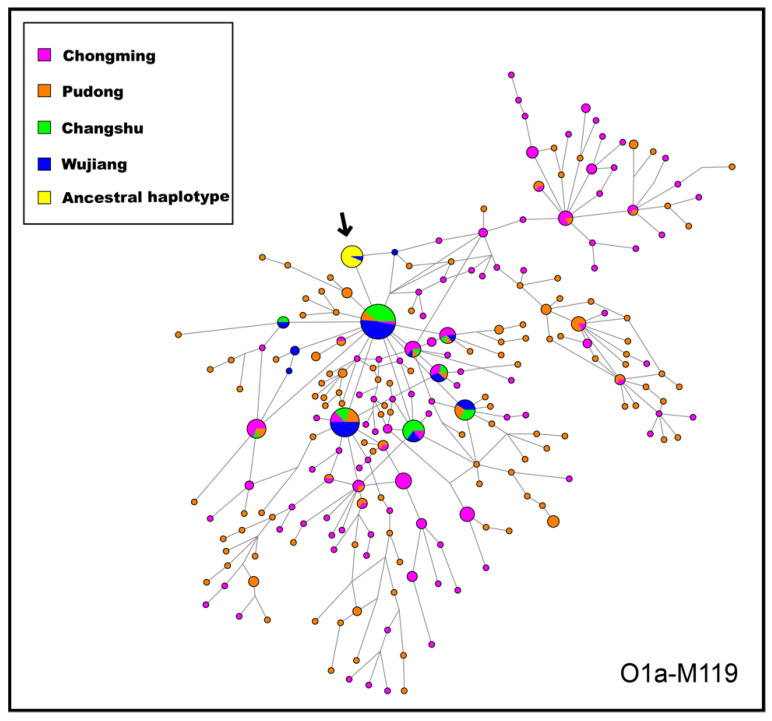
Median-joining network of Y chromosomal short tandem repeat (Y-STR) haplotypes of four populations of haplogroup O1a-M119.

**Table 1 genes-13-01363-t001:** Calculation of forensic parameters from 4 different Y-STR genotyping systems.

Time(s) a Haplotype Was Observed	Y Filer		PPY23		Y Filer Plus		Y Filer Platinum	
	PD	CM	PD	CM	PD	CM	PD	CM
1	573	388	630	462	657	508	669	522
2	30	27	26	15	16	8	10	4
3	9	9	1	6		2		
4	6	3	1	2				
5	1	3		1				
6		3						
7		1		1				
9		1						
No. of haplotypes	619	435	658	487	673	518	679	526
FUH	0.83164	0.73208	0.91437	0.87170	0.95356	0.95849	0.97097	0.98491
HD	0.99957	0.99854	0.99985	0.99946	0.99993	0.99990	0.99996	0.99997
MP	0.00189	0.00334	0.00160	0.00243	0.00152	0.00199	0.00149	0.00192
DC	0.89840	0.82075	0.95501	0.91887	0.97678	0.97736	0.98549	0.99245

PD: Pudong Population (*n* = 689). CM: Chongming Population (*n* = 530). HD: haplotype diversity. MP: match probability. DC: discrimination capacity. FUH: fraction of unique haplotype. Yfiler: AmpFISTR^®^ Yfiler kit (17 Y-STR loci). PPY23: PowerPlex^®^ Y23 System (23Y-STR loci). YfilerPlus: AmpFISTR^®^ Yfiler Plus kit (27 Y-STR loci). Yfiler™ Platinum PCR Amplification Kit (38 Y-STR LOC).

## Data Availability

Data were available within the article or its Appendix A.

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
