# Peer review of "Forensic Analysis and Genetic Structure Construction of Chinese Chongming Island Han Based on Y Chromosome STRs and SNPs"

_genes, 2022, doi:10.3390/genes13081363_

Round 1
Reviewer 1 Report
This is a review of the manuscript “Forensic Analysis and Genetic Structure Construction of Chinese Chongming Island Han based on Y chromosome STRs and SNPs” that was submitted for publication if Genes (genes-1708891) by Shilin Li and co-authors.
In this manuscript, the authors present the results of a Y-STR and Y-SNP screening of a number of male groups from eastern China. The authors compare the obtained results with those from previous relevant studies and present their results in some detail.
I have a large number of comments and remarks which I will try to list below.
1. The authors should very carefully completely check all reference citations in this manuscript. Starting at least from reference 46 onwards, there is at least one shift, but there might be more.
2. The authors should also check citations of Figures. E.g. on line 210 (page 5) there should be a citation to Figure 2 and on line 26 (page 8) there should be a citation to Figure 3 (but there could be more!).
3. The first citations to references should be revised. Reference 1 should be replaced by one or more of the Nature Review Genetics Y-chromosome reviews of Mark Jobling and others. Reference 2 does not make any sense at all. Reference 3 does not discuss in detail the different Y-chromosome mutation rates, and I could go on and on. It is fine to cite some studies by Chinese authors where relevant (e.g. sources of Chinese populations), but none of these were innovative and are preceded by many much more suitable articles published by western scientists. For an unbiased history of the use of the human Y-chromosome, see https://www.mdpi.com/2073-4425/13/5/898/htm, and carefully extract relevant publications from that. As an example, on line 65 (of page 2), the sentence should end with a reference to article number 13 of that manuscript. This article, by the Knijff et al, is the first to discuss the differential use of Y-STRs and Y-SNPs for various purposes. Proper citing should always refer to original sources, not to much more recent articles by friends.
4. lines 79 to 92 (of page 2) partially contains results and should be moved to another location in the manuscript. Furthermore, it is not surprising that a multiplex Y-STR kit as Y-Filer Platinum has a higher discriminating power, it contains more loci than other kits!
5. re line 98 (page 3), how was unrelatedness established without compromising the privacy rights of the sample donors? Was there a possibility of study subjects to object against participation and could they refuse participation? Can this be proven, as such should be included in the informed consent forms?
6. re. line 110 – 114 (page 3). Here, the authors list all Y-STR loci in the platinum kit, but they do not present a list of the number of loci in the others Y-STR kits they use (see Table 1). It would be wise to do this in a supplementary table, and delete this list here.
Furthermore, on line 111, they list DYS389I and DYS389II but, e.g. on line 183 (page 5), they use DYS389b. Please explain. Furthermore, as DYS389I is part of 389II, before the use of these two loci the genotypes should be adjusted. Please explain how the authors dealt with this (see for its primary source: https://www.nature.com/articles/5200223).
7. Please give the source of the Gene Diversity (line 136, page 4).
8. The formula given between lines 140 and 141 (page 4) shows GH/HD, please explain?
9. re. line 163 (page 4), for use in MDS, not Rst, but linearized Rst should be used in other to avoid inclusion of negative values or values over 1.
10. re line 193 (page 5), insert ‘locus’ between ‘STR’ and ‘are’.
11. re. line 23 (page 8), what do the authors mean with ‘deeply verify’?
12. What do the authors mean with ‘geographically bordered’? (40, page 9).
13. Please delete the NJ tree (figure 4) and its discussion from the text. This figure does not add anything significant, especially because it was not bootstrapped in order to reveal any strong or weakly supported branching.
14. For the Y-SNP part of the manuscript, the authors should (1) present a tree with the hierarchical ordering of all the SNPS they used and the each branch labelled with the inferred Y (sub)haplogroup. In a separate table, these haplogroups can be listed together with all relevant number of observations and frequencies in the populations tested. Subsequently, the present figure 5 can be deleted.
15. re line 158 (page 13), it is median joining, not join.
16. reference 65 (line 164, ,page 13) does not explain how to infer the ancestral haplotype but discusses genome wide structural DNA variation. Please provide the correct source of identifying an ancestral haplotype, explain how this works, and do this in the methods section of this manuscript.
Author Response
Dear Reviewer:
On behalf of my co-authors, we thank you very much for giving us an opportunity to revise our manuscript, we appreciate very much for your positive and constructive comments and suggestions on our manuscript entitled “Forensic Analysis and Genetic Structure Construction of Chinese Chongming Island Han based on Y chromosome STRs and SNPs” (Manuscript ID: genes-1708891).Those comments are all valuable and very helpful for revising and improving our paper, as well as the important guiding significance to our researches. We have studied comments carefully and have made correction which we hope meet with approval. We have made a World file to show our response as the attachment.
Thank you again for your kind suggestions for helping us improve our manuscript.

Reviewer 2 Report
An interesting paper - only a few minor coments from me
Line 34: Rephrase "were carried outperformed": does not read well
Line "results showed revealed"
Figure 2: Explain different colours for points. And expand the legend "Boxplots comparing GD values…"
Line 238: What is the difference between "highest" and "strongest" discriminative abilities? Only one term is needed.
Line 26: Shouldn't this be figure 3? Why are the initial stress levels different in the fig (0.0573 and 0.04649)
Line 51: rephrase: The Chongming island population is the result of immigration of the mainland population. Since transportation between the island and the mainland is difficult, this has led to partial geographic isolation, resulting in genetic drift.
Legend Fig 4: The illustrations correspond to those shown in figs 1,3
Line 165: Fig 7?
Author Response
Dear Reviewer:
On behalf of my co-authors, we thank you very much for giving us an opportunity to revise our manuscript, we appreciate very much for your positive and constructive comments and suggestions on our manuscript entitled “Forensic Analysis and Genetic Structure Construction of Chinese Chongming Island Han based on Y chromosome STRs and SNPs” (Manuscript ID: genes-1708891).Those comments are all valuable and very helpful for revising and improving our paper, as well as the important guiding significance to our researches. We have studied comments carefully and have made correction which we hope will meet with approval. We have made a point-by-point response in a world file. Please see the attachment.
Thank you again for your kind suggestions for helping us improve our manuscript.

Reviewer 3 Report
Over 500 male Han individuals in each of two regions, Pudong and Chongming Island, were genotyped for 38 Y STR loci and 24 Y SNPs. The haplotype diversity was >0.9999 in both regions with over 97% unique haplotypes in both groups. This large dataset has been analyzed in many ways. For a reader not familiar with all of the subgroups of Han Chinese, the detailed sentences in the descriptions of the results are difficult to follow. However, they seem OK. Overall, the extensive analyses of Y chromosome data on large numbers of males in each of the populations simply show the two populations are relatively similar to each other compared to other groups except the Jiangsu.
There are immediate problems with the English in the abstract and introduction: several sentences note that some populations are “more” similar without a following “than what?” This is a small grammatical issue but immediately leads to confusion. The comparison issue is pervasive. For example, later it is said that the “Japanese Nagasaki and Okinawa populations exhibited closer genetic affinity” without indication of what the comparison is to. The authors should go through the paper and look at each occurrence of a comparative word (e.g., closer, more) and make sure the comparison is complete.
Specific points
The legends to essentially all figures are of such small size that they are unreadable even when enlarged: too few pixels for the fine information. I assume the figures will be in higher quality in the final paper.
Lines 73-78 are a bit confusing. The migration influx into Shanghai is mentioned and then the natives of Pudong are studied. What are “floating” populations. That the two studied populations have unique genetic backgrounds is not supported by the preceding material but rather is a question to be answered by this study.
Lines 137-138 do not define the term HD in the formula.
Line 174: “reference populations”
Page 12 lines 114ff: the sentence says the “distribution ratio” was 21.54A%. Where is the ratio?
Line 145. The sentence is ambiguous. What is the “internal” structure? Internal to what? Is it in both the north and south Chinese? Or is it that the north and south Chinese are different?
Author Response
Dear Reviewer:
On behalf of my co-authors, we thank you very much for giving us an opportunity to revise our manuscript, we appreciate very much for your positive and constructive comments and suggestions on our manuscript entitled “Forensic Analysis and Genetic Structure Construction of Chinese Chongming Island Han based on Y chromosome STRs and SNPs” (Manuscript ID: genes-1708891).Those comments are all valuable and very helpful for revising and improving our paper, as well as the important guiding significance to our researches. We have studied comments carefully and have made correction which we hope meet with approval. The main corrections in the paper and the responds to your comments could be seen in the attachment.
Thank you again for help us improving our manuscript.
